# ROBUST DIRECT PREFERENCE OPTIMIZATION VIA VARIATIONAL FORM F-DIVERGENCE

## ABSTRACT

Direct Preference Optimization is commonly deployed to align Large Language Models (LLMs) with human preferences, while such a technique also suffers from noisily annotated human preference. Existing robust approaches often require the knowledge of transition between clean and noisy human preferences, or leverage additional architecture/models to perform noisy human preference correction. In this work, we investigate when $f$-divergence is immune to the imperfect human preference annotations, by maximizing the $f$-divergence between noisy preferred and unpreferred data distributions. Theoretically, we show that when the noise ratio is known, the Total Variation formulation can serve as a surrogate for the clean dataset. In contrast, the Jensen–Shannon formulation is invariant to noise, yielding identical results under both noisy and clean preferences, even without knowledge of the noise rate. Empirically, the variational form of the Jensen–Shannon divergence enhances the model's ability to generate preferred responses under noisy conditions, while simultaneously improving the factual accuracy of its outputs.

## 1 INTRODUCTION

Direct Preference Optimization (DPO) (Rafailov et al., 2023) has been established as a simple yet effective alternative to Reinforcement Learning from Human Feedback (RLHF) (Ouyang et al., 2022) for aligning Large Language Models (LLMs) (Achiam et al., 2023) with human preferences (Christiano et al., 2017). In contrast to RLHF, which relies on training a reward model with reinforcement learning methods including Proximal Policy Optimization (PPO) (Schulman et al., 2017), DPO learns a policy directly from human preference data, instead of employing explicit reward modeling. While DPO efficiently captures preference information from pairwise data via a binary cross-entropy loss (Shannon, 1948), noisy annotations can cause the policy to learn misleading or suboptimal patterns.

To mitigate the influence of noisy preference data, the authors of DPO account for known label-flipped probabilities and leverage a binary cross-entropy objective to estimate a conservative target distribution, thereby stabilizing model updates under uncertainty (Mitchell, 2023). However, overly conservative gradient updates constrain the model close to the reference policy, limiting its learning capability, and resulting in substantial performance gaps compared to the original DPO. To overcome these limitations, robust DPO (Chowdhury et al., 2024) was proposed in place of conservative DPO, providing an unbiased estimate of DPO. By introducing a multiplicative factor to counteract the effects of noisy preferences and employing dynamic gradient updates, Robust DPO enhances learning stability and model performance. Although it partially reduces the suboptimality gap relative to the optimal policy, it undeniably remains highly dependent on knowledge of the transition between clean and noisy human preferences.

Among approaches that do not require knowledge of the noise rate, two methods have shown particularly strong performance. Identity Preference Optimization (IPO) (Azar et al., 2024), while not specifically developed for robustness, leverages an identity mapping trick that stabilizes the training objective and, as a result, exhibits notable robustness in practice. However, there is no rigorous theoretical guarantee explaining how IPO functions under noisy data, and its effectiveness diminishes substantially as the noise level increases. Another method tailored for pairwise noise, Dr.DPO (Wu et al., 2024), transfers the Distributionally Robust Optimization (DRO) (Duchi & Namkoong, 2021) framework to DPO, reweighting the gradients to account for noise and improve robustness.

While Dr.DPO exhibits notable efficacy on both clean and noisy data, it primarily serves as a general framework for enhancing policy learning and lacks rigorous theoretical justification regarding its robustness to noise.

To maintain robustness and avoid reliance on noise rate estimation, we propose a Direct Preference Optimization method via $f$-divergence ($f$-DPO), which is inherently robust to noise. In particular, we leverage $f$-divergence to measure the preferred and unpreferred distributions in DPO, recasting reward maximization as maximizing the corresponding $f$-divergence. We then employ its variational form to modify the loss function, assigning optimal importance to noisy data. Our main contributions are highlighted below.

- We demonstrate how, given knowledge of the noise rate, the variational form of Total Variation under noisy conditions can be transformed into its corresponding variational form under the clean distribution.
- For a more general class of settings, we prove that the variational form of Jensen–Shannon divergence remains invariant under noisily annotated preferences, thereby eliminating the need to estimate label flipping rates.
- Extensive experiments conducted on the HH-RLHF and UltraFeedback datasets, as well as the MT-Bench and TruthfulQA benchmarks, validate the robustness of the Jensen–Shannon formulation against preference noise, while also enhancing policy reasoning and factual accuracy of model outputs.

## 2 RELATED WORKS

### 2.1 $f$-DIVERGENCE WORKS

$f$-divergences have been widely adopted in deep learning, owing to their versatile and valuable properties. Early $f$-GANs (Nowozin et al., 2016) employed f-divergences as the training objective for generative adversarial networks(GANs), and were later adapted in DPO to characterize the discrepancy between the recent policy and the reference policy (Wang et al., 2023). In contrast, our method of introducing f-divergences is designed to maximize the distinction between the distributions of preferred and unpreferred responses.

The earliest connection between $f$-divergences and robust training was established in the context of classification tasks (Wei & Liu, 2020; Novello & Tonello, 2024). Subsequently, $f$ divergence was incorporated as a means to enhance Weak-to-Strong Generalization for large language models (Yao et al., 2025).

### 2.2 ROBUST DPO WORKS

Initially, cDPO (Mitchell, 2023) addresses the bias induced by label flipping by employing label smoothing. Subsequently, rDPO (Chowdhury et al., 2024) is designed as an unbiased loss function, by explicitly modeling the stochastic flip rate of preference labels. Additionally, the GRPO (Ramesh et al., 2024) method, built upon the reward-free direct preference optimization framework, utilizes alternating optimization and mirror descent to iteratively update population weights and the policy, aiming to minimize the worst-case loss. Later, the Dr.DPO (Wu et al., 2024) framework was formulated to strengthen robustness by optimizing preference pairs under worst-case scenarios via Distributionally Robust Optimization. Building on the DRO paradigm, a recent framework (Xu et al., 2025) introduces a min–max loss formulation to robustly optimize against the worst-case distribution within the defined uncertainty set.

To the best of our knowledge, ours is the first work to investigate how the properties of $f$-divergence can enhance the robustness of Direct Preference Optimization.

## 3 PRELIMINARIES

In this section, we present the preliminary formulation of $f$-DPO. Our exposition proceeds in two stages: we first define the notation and derivation for DPO, and then provide the relevant background on $f$-divergence.

### 3.1 DIRECT PREFERENCE OPTIMIZATION

**Preference Data.** In DPO, preference dataset typically consists of human-annotated pairwise responses, denoted as $\mathcal{P} = \{(x, y_w, y_\ell)\}$, where $x$ is the input prompt, $y_w$ the preferred (chosen) response, and $y_\ell$ the unpreferred (rejected) response.

**Supervised Fine-Tuning.** As a preliminary stage for DPO, supervised fine-tuning (SFT) adjusts the model parameters by maximizing the likelihood of the target responses $y_w$, producing the reference model that serves as the foundation for subsequent DPO optimization,

$$\mathcal{L}_{SFT,\beta} = -\log \pi_\theta(y \mid x).$$

**Direct Preference Optimization.** Given a preference pair $(x, y)$, the reward in DPO is computed as the log-difference between the current policy $\pi_\theta$ and the SFT policy $\pi_{ref}$, augmented with a normalization term $Z(x)$. Formally,

$$Z(x) = \sum_y \pi_{\text{ref}}(y \mid x) \exp\big(\beta\, r(x, y)\big),$$

which serves as a partition function to guarantee that the induced probability distribution remains properly normalized. The complete reward function is defined as:

$$r(x, y) = \beta \log \frac{\pi_\theta(y|x)}{\pi_{ref}(y|x)} + \beta \log Z(x).$$

Based on the Bradley-Terry (Bradley & Terry, 1952) model for the probability of the preferred response being selected, the DPO loss function is formulated as:

$$\mathcal{L}_{DPO,\beta}(\theta) = -\log \sigma \left( \beta \log \frac{\pi_\theta(y_w|x)}{\pi_{ref}(y_w|x)} - \beta \log \frac{\pi_\theta(y_l|x)}{\pi_{ref}(y_l|x)} \right),$$

where $\sigma(\cdot)$ is the sigmoid fuction, and the temperature parameter $\beta$ governs the degree of exploration in the policy, effectively controlling the confidence of the model in distinguishing preferred from less-preferred responses.

### 3.2 $f$-DIVERGENCE

Our method is built upon the formulation of $f$-divergence, leveraging its properties to design a robust preference optimization framework. $f$-divergence is a broad class of measures for quantifying the discrepancy between probability distributions, generalizing the concept of KL divergence. Formally, for two probability distributions $P$ and $Q$:

$$D_f(P\|Q) = \int Q(x)\, f\left(\frac{P(x)}{Q(x)}\right) dx,$$

where $f(\cdot)$ is a convex function satisfying $f(1) = 0$. As illustrative examples, we place particular emphasis on JS divergence, which corresponds to $f(u) = \frac{1}{2}\left[u \log u - (u + 1) \log \frac{1+u}{2}\right]$, and TV divergence, defined by $f(u) = \frac{1}{2}|u - 1|$. Leveraging Fenchel's convex duality, $f$-divergence admits the following variational formulation:

$$D_f(P\|Q) = \sup_{g:\mathcal{Z}\to\mathrm{dom}_{f^*}} \mathbb{E}_{\mathcal{Z}_p \sim P}[g(\mathcal{Z}_p)] - \mathbb{E}_{\mathcal{Z}_q \sim Q}[f^*(g(\mathcal{Z}_q)],$$

where $f^*$ denotes the Fenchel conjugate of $f(\cdot)$, defined as $f^*(u) = \sup_v \{uv - f(u)\}$, while $dom_{f^*}$ indicates the domain of $f^*$. To ensure that the expectations and variances in the theoretical derivations are bounded, the ratio between the two probability density functions $\mathcal{Z}_p/\mathcal{Z}_q$ is required to remain within a controlled range, implying that both distributions share the same support without extreme deviations (Suzuki et al., 2008).

The optimal function $g^*$ associated with $D_f(P\|Q)$ can be obtained directly from the ratio of the probability densities in the integral definition of the $f$-divergence. Table 1 summarizes the explicit forms of the different divergence measures.

$$g^* = \arg\sup_g\{D_f(P\|Q)\} = f'\left(\frac{P(x)}{Q(x)}\right). \tag{1}$$

By substituting the optimal function associated with each $f$-divergence, $D_f$ can be expressed in the following form:

$$D_f(P\|Q) = \mathbb{E}_{\mathcal{Z}_p \sim P}[g^*(\mathcal{Z}_p)] - \mathbb{E}_{\mathcal{Z}_q \sim Q}[f^*(g^*(\mathcal{Z}_q))],$$

which serves as the foundation for developing our robust objectives in preference optimization.

## 4 $f$-DPO: ROBUST WITH PREFERENCE NOISE

In this section, we introduce $f$-divergence to reformulate the DPO loss function, leveraging its variational form to align the model with preferred responses and away from unpreferred responses. To derive our method, we first express the optimization objective as the $f$-divergence between two distributions, and then introduce variational form to replace direct maximization of the $f$-divergence. In section § 4.2, we explore the mechanisms by which robustness is preserved under scenarios with both known and unknown noise ratios.

### 4.1 MAXIMIZE $f$-DIVERGENCE TO ADJUST LOSS FUCTION

We begin by considering a preference dataset $\mathcal{P} = \{(x, y_w, y_\ell)\}$, where $y_w$ and $y_l$ denote the chosen and rejected responses, respectively. DPO fine-tunes the original model on $\mathcal{P}$ to better align with preferred responses, enforcing a KL divergence constraint to maintain proximity to the reference model, where $\pi_\theta$ denotes the training policy and $\pi_{ref}$ denotes the reference policy. The preference framework is then instantiated using $log\sigma$, the log-sigmoid fuction, from the Bradley-Terry model, as follows:

$$\mathcal{L}_{DPO,\beta}(\theta) = -\log \sigma \left( \beta \log \frac{\pi_\theta(y_w|x)}{\pi_{ref}(y_w|x)} - \beta \log \frac{\pi_\theta(y_l|x)}{\pi_{ref}(y_l|x)} \right),$$

where $h(\pi_\theta, \pi_{\theta_{ref}}, y) = \log \frac{\pi_\theta(y|x)}{\pi_{ref}(y|x)}$, the logarithmic probability ratio of tokens generated by $\pi_\theta$ relative to $\pi_{\theta_{ref}}$, can be considered as the pointwise KL divergence from a generalized perspective. It is straightforward to see that minimizing $L_{DPO}$ is equivalent to maximizing $h(\pi_\theta, \pi_{\theta_{ref}}; y_w) - h(\pi_\theta, \pi_{\theta_{ref}}; y_\ell)$. For our $f$-DPO, we leverage $f_{div}(D_{KL|w}\|D_{KL|\ell})$ to quantify the divergence between the chosen and rejected sets. Thus, the ultimate optimization objective reduces to deriving $\pi_\theta$ under which $f$ satisfies the upper bound, i.e., $\pi_\theta = argmax_\theta f_{div}(D_{KL|w}\|D_{KL|\ell})$.

**Variational Formulation.** For more tractable computation, $f$-divergence is reformulated via its variational representation, which allows us to maximize the associated variational gap. Notably, this variational representation should be viewed as an empirical estimate rather than a strict equivalence (Wei & Liu, 2020).

$$f(D_{KL|w}|D_{KL|\ell}) = \sup_g \left[ \mathbb{E}_{Z_w \sim D_w}[g(Z_w)] - \mathbb{E}_{Z_\ell \sim D_\ell}[f^*(g(Z_\ell))] \right] = \sup_g \text{VF}(\theta, g), \quad (2)$$

here, in the equation, $f^*$ is defined as the conjugate function of the $f$-divergence function, $D_w$ denotes the preferred distribution, $Z_w = h(\theta, \theta_{ref}, y_w)$ and $Z_\ell = h(\theta, \theta_{ref}, y_\ell)$. Specifically, the former is drawn from the distribution of the chosen set, whereas the latter is drawn from the distribution of the rejected set.

We denote $g^*$ as the function that maximizes the variational objective as Eqn. (2). For various divergences, the corresponding forms of $g^*$ and its transformatio $f^*(g)$ are depicted in Table 1 (Wei & Liu, 2020; Nowozin et al., 2016). Given access to a set of preference data $(x, y_w, y_\ell)$, the per-sample maximized variational function can be expressed as follows:

$$\sup_g \text{VF}(\theta, g) = \text{VF}(\theta, g^*) = g^*(Z_w) - f^*(g^*(Z_\ell)).$$

**The modified loss $L_{f,\beta}(\theta)$.** Consequently, the loss function of $f$-DPO for any given pair of preference data can be expressed as:

$$\mathcal{L}_{f,\beta}(\theta) = -\log \sigma \big( \beta(\text{VF}(\theta, g^*)) \big). \quad (3)$$

Table 1: optimal variational $g$ ($g^*$), conjugate functions($f^*$)

| Name | $g^*(v)$ | $\mathrm{dom}_{f^*}$ | $f^*(u)$ |
|---|---|---|---|
| Total Variation | $\dfrac{1}{2}\tanh v$ | $u \in \left[-\frac{1}{2}, \frac{1}{2}\right]$ | $u$ |
| Jensen-Shannon | $\log \dfrac{2}{1+e^{-v}}$ | $u < \log 2$ | $-\log(2-e^u)$ |
| Pearson | $v$ | $\mathbb{R}$ | $\dfrac{1}{4}u^2 + u$ |
| KL | $v$ | $\mathbb{R}$ | $e^{u-1}$ |

## 4.2 WHEN $f$-DPO IS ROBUST WITH PREFERENCE NOISE

**Noise defination.** Considering pairwise noise, let $e_w$ denote the proportion of chosen responses that are flipped to rejected, while $e_\ell$ denotes the opposite. In general, we assume that the flipping noise occurs in pairs, i.e., $e_w = e_\ell$. Accordingly, we refer to the paired flipping noise $e_f$ in the following. More precisely, this can be represented as transforming $(x, y_w, y_\ell)$ into $(x, y_\ell, y_w)$ within the dataset $\mathcal{P}$ with probability $e_f$

**Noisy set.** For noisy datasets $\widetilde{\mathcal{P}} = \{(x, \widetilde{y}_w, \widetilde{y}_\ell)\}$, we denote the noisy distributions as $\widetilde{D}_w$ and $\widetilde{D}_\ell$. To demonstrate the behavior of $f$-DPO in the presence of noise, we introduce the following noisy variational form,

$$\widetilde{\mathrm{VF}}_f(\theta, g) = \mathbb{E}_{\widetilde{Z}_w \sim \widetilde{D}_w}[g(\widetilde{Z}_w)] - \mathbb{E}_{\widetilde{Z}_\ell \sim \widetilde{D}_\ell}[f^*(g(\widetilde{Z}_\ell))], \tag{4}$$

where the log-probability discrepancy $\widetilde{Z}_w = h(\theta, \theta_{ref}; \widetilde{y}_w)$ and $\widetilde{Z}_\ell = h(\theta, \theta_{ref}; \widetilde{y}_\ell)$.

**Connect to clean set.** Next, we describe how it can be closely connected to the variational difference terms defined on the clean distributions. formalizing a transformation that maps noisy datasets to their clean counterparts:

$$\mathbb{E}_{\widetilde{Z}_w \sim D(x, \widetilde{y}_w)}\left[g(\widetilde{Z}_w)\right] = \mathbb{E}_x\left[((1-e_w))\,g(Z_w) + e_\ell g(Z_\ell)\right]. \tag{5}$$

**Under the assumption of known noise ratio.** We now discuss the behavior of Total Variation(TV) under a known noise ratio. For pair data $(y_w, y_l)$, we define the following component: $\triangle_{TV}^{y_w}(\theta, g) = e_\ell \mathbb{E}_{Z_w \sim D_w}[g(Z_w)] - e_w \mathbb{E}_{Z_w \sim D_w}[f^*(g(Z_w))]$ and $\triangle_{TV}^{y_l}(\theta, g) = e_\ell \mathbb{E}_{Z_\ell \sim D_\ell}[g(Z_\ell)] - e_w \mathbb{E}_{Z_\ell \sim D_\ell}[f^*(g(Z_\ell))]$.

**Theorem 1.** *Consider Total Variation, the variational formulation under preference noise relates to that under the clean distribution as follows:*

$$\widetilde{VF}_{TV}(\theta, g) = (1 - 2e_f)VF_{TV}(\theta, g) + Bias_{TV}(\theta, g), \tag{6}$$

where $\mathrm{Bias}_{TV}(\theta, g) = \triangle_{TV}^{y_w}(\theta, g) + \triangle_{TV}^{y_l}(\theta, g)$. As the Fenchel conjugate $f_{TV}^*(u) = u$ , $\triangle_{TV}^{y_w}(\theta, g) \equiv 0$ and $\mathrm{Bias}_{TV} \equiv 0$. Since $\mathrm{Bias}_{TV}$ is negligible, the optimization objective under noise can be effectively mapped to that under the clean distributions with a multiplicative factor $(1 - 2e_f)$.

**Theorem 2.** *Under the knowledge of transition between clean and noisy human preferences, total variation is robust via its variational form.*

$$\widetilde{VF}_{TV}(\theta, g) = (1 - 2e_f)VF_{TV}(\theta, g). \tag{7}$$

**Proof Sketch.** By examining Eqn. (5) and its form under the conjugate function $f^*$, it is observed that additional terms corresponding to the distributions $D_w$ and $D_\ell$, respectively, can be incorporated separately. For the former part,

$$\mathbb{E}_{\widetilde{Z}_w \sim D(x, \widetilde{y}_w)}\left[g(\widetilde{Z}_w)\right] = \mathbb{E}_x\left[(1 - e_w - e_\ell)\,g(Z_w) + e_\ell g(Z_l) + e_\ell g(Z_w)\right],$$

and the latter part is handled in the same manner. Then, by combining the two resulting expressions, we obtain Eqn. (6). Under this premise, our bias term can be interpreted as the sum of two variational

formulations, each constructed such that the preceding and succeeding terms of $f$-divergence variational form are taken w.r.t. to the same distribution. This implies that when the conjugate function $f^*$ is directly substituted into the bias term, the resulting value is identically zero. Consequently, for Total Variation, $\widetilde{VF}_{TV}(\theta, g)$ can be explicitly transformed into $VF_{TV}(\theta, g)$ through the noise rate $e_f$. $\qquad\square$

**General setting.** Since accurately measuring noise in a preference dataset is often challenging, we next show how Jensen–Shannon (JS) divergence reduces dependence on explicit noise estimates.

**Theorem 3.** *Analogous to Total Variation, the variational formulation of Jensen–Shannon divergence satisfies the following relationship.*

$$\widetilde{VF}_{JS}(\theta, g) = VF_{JS}(\theta, g) + Bias_{JS}(\theta, g). \tag{8}$$

where $\text{Bias}_{JS}(\theta, g) = \triangle_{JS}^{y_\ell}(\theta, g) - \triangle_{JS}^{y_w}(\theta, g)$, while $\triangle_{JS}^{y_\ell}(\theta, g) = e_\ell \mathbb{E}_{Z_\ell \sim D_\ell}[g(Z_\ell)] - e_w \mathbb{E}_{Z_w \sim D_w}[f^*(g(Z_w))]$ and $\triangle_{JS}^{y_w}(\theta, g) = e_w \mathbb{E}_{Z_w \sim D_w}[g(Z_w)] - e_\ell \mathbb{E}_{Z_l \sim D_\ell}[f^*(g(Z_\ell))]$.

When computing the loss, we maximize the variational objective over $g$. Formally, letting $g^* = \arg\sup_g \{D_f(P\|Q)\}$ as Eqn. (1). Given this assumption, the bias term in the variational formulation of Jensen–Shannon divergence can be written as follows:

$$\text{Bias}_{JS}(\theta, g^*) = e_f[f_{JS}(D_\ell\|D_w) - f_{JS}(D_w\|D_\ell)].$$

Jenshon-Shannon is inherently symmetric. Unlike asymmetric KL divergence, Jensen–Shannon divergence is defined as the average of two KL divergences evaluated w.r.t. their mixture distribution. Specially,

$$f_{JS}(D_w\|D_\ell) = f_{JS}(D_\ell\|D_w) = \frac{1}{2}D_{KL}(D_w\|M) + \frac{1}{2}D_{KL}(D_l\|M),$$

where $M = \frac{1}{2}(D_w + D_\ell)$, which endows Jensen–Shannon divergence with a symmetric and bounded distance measure, taking values within the range $[0, log2]$. It is evident that the bias term of the JS variational formulation can be eliminated. In other words, the variational formulation under noisy preference is directly equivalent to that under the clean distributions for Jenshon-Shannon.

**Theorem 4.** *Under the assumption of the optimal $g^*$, the Jensen–Shannon formulation remains invariant to preference noise.*

$$\widetilde{VF}_{JS}(\theta, g^*) = VF_{JS}(\theta, g^*). \tag{9}$$

**Proof Sketch.** Let $VF_{JS}(\theta, g)$ denote the variational form of Jensen–Shannon divergence between the preferred distribution $D_w$ and the unpreferred distribution $D_\ell$, and $VF_{JS}^*(\theta, g)$ denote its symmetric form. According to Eqn. (2), $f$-DPO seeks parameters that maximize the upper bound of the variational form, expressed as $\sup_g VF(\theta, g) = VF(\theta, g^*)$. Under this assumption, Eqn. (8) can be correspondingly simplified to,

$$\widetilde{VF}_{JS}(\theta, g^*) = VF_{JS}(\theta, g^*) + \text{Bias}_{JS}(\theta, g^*).$$

while its symmetric form undergoes an analogous transformation. Consequently, the original bias term can be reformulated as a difference of symmetric JS divergences. By the inherent symmetry of JS divergence, the bias term is eliminated, which confirms that the JS variational form remains invariant under flipping noise label. $\qquad\square$

By virtue of Theorem 2 and Theorem 4, we formulate two robust variational objectives. Since the JS variational formulation in Eqn. (9) does not require prior knowledge of the noise rates, it is better suited for generalization. Accordingly, we focus on evaluating the performance of the JS variational form in our experiments, with its corresponding loss function being considered proportional to Eqn. (3):

$$\widetilde{\mathcal{L}}_{JS,\beta}(\theta) \propto \mathcal{L}_{JS,\beta}(\theta) = -\log \sigma\big(\beta(VF_{JS}(\theta, g^*))\big). \tag{10}$$

## 5 EXPERIMENTS

In this section, we conduct a series of experiments to evaluate the robustness of our method, while examining the impact of $f$-DPO on reasoning ability and output fidelity under preference noise,

and finally exploring its potential application across different tasks, models, and datasets. Initially, we assess the robustness of $f$-DPO in maintaining stable training under noisy conditions using the Anthropic HH-RLHF (Bai et al., 2022) dataset, where it surpasses existing baselines. Subsequently, we perform comprehensive evaluations on the UltraFeedback (Cui et al., 2023) dataset, where our approach consistently achieves superior performance. Building on this, we further extend our evaluation to two benchmarks: MT-Bench (Zheng et al., 2023) and TruthfulQA (Lin et al., 2021). Unless stated otherwise, all experiments in this work employ Jensen-Shannon divergence as the $f$-divergence metric.

**Baselines.** We present a comparative study of existing DPO variants, as outlined below: (i).The original DPO, a newly introduced, streamlined method for aligning language models with human preferences; (ii).Conservative DPO (cDPO) (Mitchell, 2023), a variant introduces $\varepsilon$ as noise and employs binary cross-entropy loss to diminish the model's confidence in incorrect preference labels; (iii)Identity Preference Optimization(IPO) (Azar et al., 2024), an alternative avoids the Bradley-Terry modeling assumption by employing an arbitrary non-decreasing mapping $\psi$, relying entirely on pairwise preference expressions; (iv)Provably Robust DPO(rDPO) (Chowdhury et al., 2024), an unbiased estimator of DPO, mitigates label flip noise by calibrating the label flip rate $\varepsilon$ and applying importance-weighted gradient updates. (v)Distributionally robustfying DPO(Dr.DPO) (Wu et al., 2024), incorporates a distributionally robust optimization (DRO) framework to explicitly optimize for the worst-case pairwise distribution, and introduces a hyperparameter $\beta'$ to control the influence of noisy data.

## 5.1 ROBUSTNESS OF $f$-DPO ON HH-RLHF

In this section, we conduct experiments on Anthropic HH-RLHF, a multi-turn dialogue preference dataset, where each example pairs chosen and rejected assistant responses for the same prompt, designed to train reward models that align language models with both helpfulness and harmlessness. To evaluate robustness under varying noise levels, we introduce pairwise label-flipping with flip rates of $10\%$, $20\%$, $30\%$, and $40\%$ across all experiments, using the Pythia-2.8B (Biderman et al., 2023) model.

**Metrics. Preference Accuracy** and **Win-Loss Rate**. **Preference Accuracy**, defined as the probability that the reward for the chosen response exceeds that of the rejected response, is measured as the proportion of test instances in the Anthropic HH-RLHF dataset where $r(x, y_w) > r(x, y_l)$. To more rigorously evaluate our approach, we conduct pairwise comparisons against the baselines with $20\%$ noisy label on the MT-Bench (Zheng et al., 2023). It is specifically designed to assess LLMs in multi-turn conversations, aiming to measure their coherence, informativeness, and interactive capabilities, while using GPT-4 as the judge. The comparison outcome, reported as **Win-Loss Rate**, consists of three components: win, loss, and tie rate.

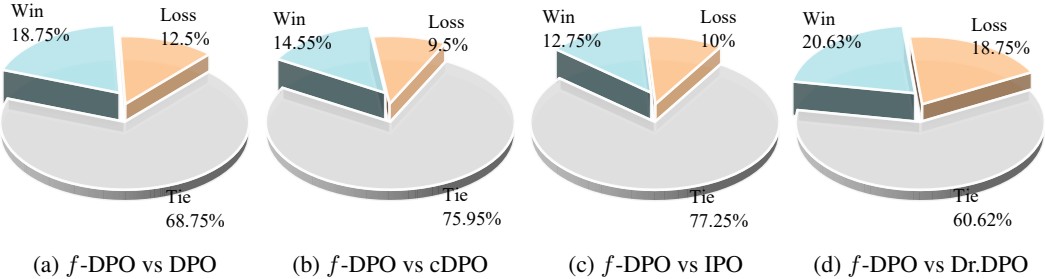

|  (a) $f$-DPO vs DPO  |  (b) $f$-DPO vs cDPO  |  (c) $f$-DPO vs IPO  |  (d) $f$-DPO vs Dr.DPO  |

Figure 1: **Win-Loss Rate** on MT-bench.

**Well performance accross varing noisy levels.** With the addition of noise, training on HH-RLHF tends to be unstable, and the maximum preference accuracy is often not reached at the final step. In this case, we report the peak preference accuracy evaluation throughout the training process. Under different levels of flipped noise, $f$-DPO consistently attained the highest preference accuracy, outperforming DPO with improvements ranging from $2.83\%$ to $4.69\%$, as reported in Table 2. It is worth noting that all baselines adopt symmetric noise in their papers. Among the baselines, IPO and Dr.DPO performed relatively well. However, IPO's performance deteriorates noticeably as the

Table 2: **Preference Accuracy** under different methods

| Noise rate | DPO | cDPO | rDPO | IPO | DrDPO | $f$-DPO |
|---|---|---|---|---|---|---|
| $e_f$=0.1 | 62.11 | 62.67 | 63.28 | 65.23 | 65.38 | 66.02 |
| $e_f$=0.2 | 62.01 | 61.83 | 62.53 | 64.06 | 64.19 | 64.84 |
| $e_f$=0.3 | 58.20 | 58.98 | 61.33 | 60.94 | 62.65 | 62.89 |
| $e_f$=0.4 | 55.47 | 55.86 | 57.11 | 57.81 | 58.83 | 59.77 |

noise level increases, whereas Dr.DPO is highly sensitive to the choice of $\beta^*$ and requires careful hyperparameter tuning.

$f$-**DPO outperforms the Baselines.** The experimental parameters, including temperature and max-tokens, are kept consistent with the original MT-Bench settings (Zheng et al., 2023). In pairwise response evaluations, Figure 1 illustrates that $f$-DPO significantly enhances reasoning ability on MT-Bench, delivering a $6.25\%$ performance gain over DPO.

## 5.2 Additional comparison on TruthfulQA

To verify the generalization of our method across models, datasets, and techniques, we further evaluate it on TruthfulQA (Lin et al., 2021) using LLaMA-2-7B (Touvron et al., 2023) training on the UltraFeedback dataset with $10\%$, $20\%$, $30\%$ and $40\%$ flipped label noise. UltraFeedback, is a large-scale preference dataset with over 64k instances across diverse domains. Unlike Anthropic HH-RLHF, it provides fine-grained annotations beyond binary labels, offering richer supervision for evaluating and training alignment methods. TruthfulQA, a benchmark of 817 adversarially designed questions across 38 categories, comprising three tasks, assesses the truthfulness and robustness of the model against generating misleading responses. Comprehensive comparisons of the model outputs are presented in the appendix D for reference.

$f$-**DPO enhances the factuality of responses.** In our experimental design, we adopt the mc1_targets task(Single-ture) and report results averaged over three independent runs. As seen in Table 3, $f$-DPO exhibits a clear advantage in Multiple-Choice accuracy, which indicates that our approach strengthens the factual accuracy and reliability of the model's responses. Our approach continues to exhibit robust performance at a noise level of $40\%$, outperforming alternative methods.

Table 3: Multiple-Choice Accuracy on TruthfulQA

| Noise rate | DPO | cDPO | rDPO | IPO | DrDPO | $f$-DPO |
|---|---|---|---|---|---|---|
| $e_f$=0.1 | 31.13 | 32.03 | 33.04 | 34.19 | 33.25 | 35.19 |
| $e_f$=0.2 | 30.17 | 30.34 | 31.73 | 32.15 | 31.73 | 34.09 |
| $e_f$=0.3 | 29.87 | 30.02 | 31.21 | 31.12 | 31.50 | 33.78 |
| $e_f$=0.4 | 29.50 | 29.70 | 30.84 | 30.72 | 31.30 | 33.29 |

## 5.3 Robustness of $f$-DPO with different $f$-divergence on Ultrafeedback

In this section, we investigate the impact of different divergences on DPO under label noise, as illustrated in Table 1. The experimental setup, including the model and dataset, is consistent with Section § 5.2. In all cases, the loss functions adhere to Eqn. (10), except for the Total Variation loss, which follows Eqn. (7) with flipping noise $e_f$.

As demonstrated in Figure 4, the formulation of Jensen–Shannon divergence exhibits a markedly stronger ability to discriminate response quality than the other considered $f$-divergences. This demonstrates that our method is grounded in the inherent properties of the Jensen–Shannon divergence, and that other divergences cannot readily inherit the same level of robustness. Notably, Total Variation exhibits subpar performance. This can be attributed to the fact that, when label-flipping noise is injected, the inherent noise already present in the original dataset interferes, resulting in an actual noise level that does not exactly match the target $e_f$ (Zhu et al., 2023). Given that

Jensen–Shannon divergence constitutes a symmetrized and smoothed variant of the KL divergence, KL divergence displays a similar trend, yet consistently underperforms relative to Jensen–Shannon divergence. However, all the $f$-divergences ultimately surpass the original DPO in the final step.

Table 4: Various $f$-divergences on Ultrafeedback

| **Preference Accuracy** | DPO | JS | KL | TV | Pearson |
|---|---|---|---|---|---|
| Best Accuracy | 70.70 | 75.00 | 70.70 | 70.31 | 71.48 |
| Last Accuracy | 67.58 | 75.00 | 69.92 | 69.92 | 71.48 |

## 5.4 ABLATION STUDIES ON ULTRAFEEDBACK

In this section, we perform ablation studies on the temperature parameter $\beta$ and batch size, followed by a theoretical analysis of the experimental results. The experimental configuration, encompassing both the model and dataset, follows the setup described in Section § 5.2.

**Impact of the Temperature Parameter $\beta$ in $f$-DPO.** Table 5 reports the effect of varying $\beta$ on the model's preference accuracy on UltraFeedback under 20% flipped noise. From a theoretical perspective, the temperature parameter $\beta$ controls the policy's confidence in the reward accuracy for DPO: a larger $\beta$ corresponds to higher confidence, whereas a smaller $\beta$ results in more conservative gradient updates. The experimental results indicate that both excessively high or low values of $\beta$ negatively affect the model's learning capability, for both DPO and $f$-DPO methods. This can be attributed to f-DPO inheriting the temperature parameter $\beta$ from DPO, where it plays the same role. Consequently, previous experiments have typically adopted $\beta = 0.1$ to achieve optimal performance.

<table>
<tr><td colspan="4">Table 5: Impact of $\beta$</td><td colspan="4">Table 6: Effect of Batch Size</td></tr>
</table>

| Noise rate | $\beta$ | DPO | $f$-DPO |
|---|---|---|---|
| | 0.1 | 70.70 | 75.00 |
| $e_f = 0.2$ | 0.5 | 69.53 | 74.61 |
| | 0.02 | 68.75 | 74.61 |

| Noise rate | Batch Size / lr | DPO | $f$-DPO |
|---|---|---|---|
| | 64 / 5e-7 | 70.70 | 75.00 |
| $e_f = 0.2$ | 128 / 8e-7 | 69.92 | 73.83 |
| | 32 / 3e-7 | 71.10 | 75.39 |

**Effect of Batch Size on the Training Dynamics of $f$-DPO.** In practical training regimes, modifications to the batch size generally require a corresponding adjustment of the learning rate (Smith et al., 2017). A commonly adopted principle is the linear scaling rule, which prescribes scaling the learning rate in proportion to the batch size so as to maintain a consistent per-sample gradient contribution. When employing different batch sizes in conjunction with their corresponding learning rates, the preference accuracy varies as viewed in Table 6. With excessively large batch sizes, gradients are averaged over a substantial number of samples, leading to updates that are stable yet overly conservative. This tendency pulls the model closer to the reference model, which can result in inferior performance compared to using moderate batch sizes. In contrast, smaller batch sizes amplify the influence of noise, yielding more oscillatory training dynamics. Nevertheless, they also partially enhance the capacity of the model for exploration over limited samples.

## 6 CONCLUSION

In this work, we propose $f$-DPO, by investigating the robustness of variational form $f$-divergences under noisy preference text dataset, with the objective of amplifying the distinction between the policy's responses to preferred and unpreferred human feedback. Building on the characterization of transitions between clean and noisy human preferences, we show that $f$-DPO admits a transformation between the clean and noisy regimes via Total Variation divergence. Crucially, under the general setting where noise estimation is not required, the Jensen–Shannon format $f$-DPO provides a more generalizable approach for robust training and is proven to remain invariant in the presence of flipping noise. In the absence of noise rate information and without incorporating auxiliary modules, our approach improves the fidelity of the policy's reward signals, thereby exhibiting substantial robustness under noisy preference.

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

## A  THE USE OF LARGE LANGUAGE MODELS

In this work, we employ ChatGPT-4, a classic large language model, to assist with language refinement and clarity improvement. Specifically, ChatGPT-4 is used to polish the writing style, correct grammatical errors, and enhance the overall readability of the manuscript without altering its scientific content or conclusions.

## B  FORMAL PROOF

### B.1  PROOF OF THEOREM 1: NOISY VALIATIONAL FORMULATION OF TV

$proof$. the former part of valiational formulation

$$
\mathbb{E}_{\widetilde{Z}_w \sim D(x,\widetilde{y}_w)} \left[ g(\widetilde{Z}_w) \right]
$$

$$
= h_x \left[ (1 - e_w) \, g \left( h(\pi_\theta, \pi_{\theta_{ref}}, y_w) \right) + e_\ell g \left( h(\pi_\theta, \pi_{\theta_{ref}}, y_\ell) \right) \right]
$$

$$
= \mathbb{E}_x \left[ (1 - e_w - e_\ell) \, g \left( h(\pi_\theta, \pi_{\theta_{ref}}, y_w) \right) + e_\ell g \left( h(\pi_\theta, \pi_{\theta_{ref}}, y_\ell) \right) + e_\ell g \left( h(\pi_\theta, \pi_{\theta_{ref}}, y_w) \right) \right]
$$

$$
= \mathbb{E}_x \left[ (1 - e_w - e_\ell) \, g \left( Z_w \right) + e_\ell g \left( Z_l \right) + e_\ell g \left( Z_w \right) \right]
$$

the latter part is derived as:

$$
\mathbb{E}_{\widetilde{Z}_l \sim D(x,\widetilde{y}_l)} \left[ f^* \left( g(\widetilde{Z}_l) \right) \right]
$$

$$
= \mathbb{E}_x \left[ (1 - e_\ell) f^* \left( g \left( h(\pi_\theta, \pi_{\theta_{ref}}, y_\ell) \right) \right) + e_w f^* \left( g \left( h(\pi_\theta, \pi_{\theta_{ref}}, y_w) \right) \right) \right]
$$

$$
= \mathbb{E}_x \left[ (1 - e_w - e_\ell) f^* \left( g \left( h(\pi_\theta, \pi_{\theta_{ref}}, y_\ell) \right) \right) + e_w f^* \left( g \left( h(\pi_\theta, \pi_{\theta_{ref}}, y_w) \right) \right) + e_w f^* \left( g \left( h(\pi_\theta, \pi_{\theta_{ref}}, y_\ell) \right) \right) \right]
$$

$$
= \mathbb{E}_x \left[ (1 - e_w - e_\ell) f^* \left( g(Z_l) \right) + e_w f^* \left( g(Z_w) \right) + e_w f^* \left( g(Z_l) \right) \right]
$$

We combine the two resulting expressions, $\mathbb{E}_{\widetilde{Z}_w \sim D(x,\widetilde{y}_w)} \left[ g(\widetilde{Z}_w) \right]$ and $\mathbb{E}_{\widetilde{Z}_l \sim D(x,\widetilde{y}_l)} \left[ f^* \left( g(\widetilde{Z}_l) \right) \right]$, after transformation, as follows:

$$
\widetilde{\mathrm{VF}}_{TV}(\theta, g)
$$

$$
= \mathbb{E}_{\widetilde{Z}_w \sim D(x,\widetilde{y}_w)} \left[ g(\widetilde{Z}_w) \right] - \mathbb{E}_{\widetilde{Z}_l \sim D(x,\widetilde{y}_l)} \left[ f^* \left( g(\tilde{Z}_l) \right) \right]
$$

$$
= (1 - e_w - e_\ell) \left[ \mathbb{E}_{Z_w \sim D(x,y_w)} \left[ g \left( Z_w \right) \right] - \mathbb{E}_{Z_\ell \sim D(x,y_\ell)} \left[ f^* \left( g \left( Z_\ell \right) \right) \right] \right] + Bias_{TV}(\theta, g)
$$

$$
= (1 - e_w - e_\ell) V_{D_f}(\theta, g) + Bias_{TV}(\theta, g)
$$

In which,

$$
Bias_{JS}(\theta, g)
$$

$$
= \left[ e_\ell \mathbb{E}_{Z_\ell \sim D_\ell}[g(Z_\ell)] - e_w \mathbb{E}_{Z_\ell \sim D_\ell}[f^*(g(Z_\ell))] \right] + \left[ e_\ell \mathbb{E}_{Z_w \sim D_w}[g(Z_w)] - e_w \mathbb{E}_{Z_w \sim D_w}[f^*(g(Z_w))] \right]
$$

$$
= \triangle_{TV}^{y_\ell}(\theta, g) + \triangle_{TV}^{y_w}(\theta, g)
$$

Consider $e_f = e_w = e_\ell$,

$$\widetilde{\mathrm{VF}}_{TV}(\theta, g) = (1 - 2e_f)\mathrm{VF}_{TV}(\theta, g) + \mathrm{Bias}_{TV}(\theta, g)$$

we proof the claim.

### B.2 PROOF OF THEOREM 2: ROBUSTNESS OF THE TV FORM

Given the flipping noise $e_f$, the bias term can be transformed into,

$$\mathrm{Bias}_{TV}(\theta, g)$$
$$= \triangle_{TV}^{y_\ell}(\theta, g) + \triangle_{TV}^{y_w}(\theta, g)$$
$$= e_f\Big[\mathbb{E}_{Z_\ell \sim D_\ell}[g(Z_\ell)] - \mathbb{E}_{Z_\ell \sim D_\ell}[f^*(g(Z_\ell))]\Big] + e_f\Big[\mathbb{E}_{Z_w \sim D_w}[g(Z_w)] - \mathbb{E}_{Z_w \sim D_w}[f^*(g(Z_w))]\Big]$$

As for Total Variation, $f^*(u) = u$. Then we perform the following transformation:

$$\mathrm{Bias}_{TV}(\theta, g)$$
$$= \triangle_{TV}^{y_\ell}(\theta, g) + \triangle_{TV}^{y_w}(\theta, g)$$
$$= e_f\Big[\mathbb{E}_{Z_\ell \sim D_\ell}[g(Z_\ell)] - \mathbb{E}_{Z_\ell \sim D_\ell}[g(Z_\ell)]\Big] + e_f\Big[\mathbb{E}_{Z_w \sim D_w}[g(Z_w)] - \mathbb{E}_{Z_w \sim D_w}[g(Z_w)]\Big]$$
$$= 0$$

Hence, $\mathrm{Bias}_{TV}(\theta, g) \equiv 0$. It follows that the TV form exhibits robustness.

$$\widetilde{\mathrm{VF}}_{TV}(\theta, g) = (1 - 2e_f)\mathrm{VF}_{TV}(\theta, g)$$

we proof the claim.

### B.3 PROOF OF THEOREM 3: NOISY VARIATIONAL FORMULATION OF JS

proof. Similarly to TV, first note.

$$\mathbb{E}_{\widetilde{Z}_w \sim D(x, \widetilde{y}_w)}\Big[g(\widetilde{Z}_w)\Big]$$
$$= \mathbb{E}_x\Big[((1 - e_w))\, g\left(h(\pi_\theta, \pi_{\theta_{ref}}, y_w)\right) + e_\ell g\left(h(\pi_\theta, \pi_{\theta_{ref}}, y_\ell)\right)\Big]$$
$$= \mathbb{E}_x\Big[g\left(h(\pi_\theta, \pi_{\theta_{ref}}, y_w)\right) + e_\ell g\left(h(\pi_\theta, \pi_{\theta_{ref}}, y_\ell)\right) - e_w g\left(h(\pi_\theta, \pi_{\theta_{ref}}, y_w)\right)\Big]$$
$$= \mathbb{E}_x\Big[g\left(Z_w\right) + e_\ell g\left(Z_\ell\right) - e_w g\left(Z_w\right)\Big]$$

contrast form:

$$\mathbb{E}_{\widetilde{Z}_l \sim D(x, \widetilde{y}_l)}\Big[f^*\left(g(\widetilde{Z}_l)\right)\Big]$$
$$= \mathbb{E}_x\Big[(1 - e_\ell)f^*\Big(g\big(h(\pi_\theta, \pi_{\theta_{ref}}, y_\ell)\big)\Big) + e_w f^*\Big(g\big(h(\pi_\theta, \pi_{\theta_{ref}}, y_w)\big)\Big)\Big]$$
$$= \mathbb{E}_x\Big[f^*\Big(g\big(h(\pi_\theta, \pi_{\theta_{ref}}, y_\ell)\big)\Big) + e_w f^*\Big(g\big(h(\pi_\theta, \pi_{\theta_{ref}}, y_w)\big)\Big) - e_\ell f^*\Big(g\big(h(\pi_\theta, \pi_{\theta_{ref}}, y_\ell)\big)\Big)\Big]$$
$$= \mathbb{E}_x\Big[g\left(Z_\ell\right) + e_\ell g\left(Z_w\right) - e_w g\left(Z_\ell\right)\Big]$$

After applying the transformations, the two expressions, $\mathbb{E}_{\widetilde{Z}_w \sim D(x, \widetilde{y}_w)}\Big[g(\widetilde{Z}_w)\Big]$ and $\mathbb{E}_{\widetilde{Z}_l \sim D(x, \widetilde{y}_l)}\Big[f^*\left(g(\widetilde{Z}_l)\right)\Big]$ are combined as follows:

$$\widetilde{\mathrm{VF}}_{JS}(\theta, g)$$

$$= \Big[ \mathbb{E}_{Z_w \sim D(x, y_w)} \Big[ g\left(Z_w\right) \Big] - \mathbb{E}_{Z_\ell \sim D(x, y_\ell)} \Big[ f^* \Big( g\left(Z_\ell\right) \Big) \Big] \Big] + \mathrm{Bias}_{JS}(\theta, g)$$

$$= \mathrm{VF}_{JS}(\theta, g) + \mathrm{Bias}_{JS}(\theta, g)$$

Where,

$$\mathrm{Bias}_{JS}(\theta, g)$$

$$= \Big[ e_\ell \mathbb{E}_{Z_\ell \sim D_\ell}[g(Z_\ell)] - e_w \mathbb{E}_{Z_w \sim D_w}[f^*(g(Z_w))] \Big] - \Big[ e_w \mathbb{E}_{Z_w \sim D_w}[g(Z_w)] - e_\ell \mathbb{E}_{Z_\ell \sim D_\ell}[f^*(g(Z_\ell))] \Big]$$

$$= \triangle_{JS}^{y_\ell}(\theta, g) - \triangle_{JS}^{y_w}(\theta, g)$$

Consider $e_f = e_w = e_\ell$,

$$\widetilde{\mathrm{VF}}_{JS}(\theta, g) = \mathrm{VF}_{JS}(\theta, g) + \mathrm{Bias}_{JS}(\theta, g)$$

we proof the claim.

### B.4 PROOF OF THEOREM 4: ROBUSTNESS OF THE JS FORM

**Notation.** We present a pair of variational formulations.

$$\mathrm{VF}(\theta, g) = \mathbb{E}_{Z_w \sim D_w}[g(Z_w)] - \mathbb{E}_{Z_\ell \sim D_\ell}[f^*(g(Z_\ell))]$$

$$\mathrm{VF}^*(\theta, g) = \mathbb{E}_{Z_\ell \sim D_\ell}[g(Z_\ell)] - \mathbb{E}_{Z_w \sim D_w}[f^*(g(Z_w))]$$

Given the pairwise noise $e_f$, the bias term can be transformed into,

$$\mathrm{Bias}_{JS}(\theta, g)$$

$$= \triangle_{JS}^{y_\ell}(\theta, g) - \triangle_{JS}^{y_w}(\theta, g)$$

$$= e_f \Big[ \mathbb{E}_{Z_\ell \sim D_\ell}[g(Z_\ell)] - \mathbb{E}_{Z_w \sim D_w}[f^*(g(Z_w))] \Big] - e_f \Big[ \mathbb{E}_{Z_w \sim D_w}[g(Z_w)] - \mathbb{E}_{Z_\ell \sim D_\ell}[f^*(g(Z_\ell))] \Big]$$

With respect to the loss function defined earlier, we maximize the variational functional $\mathrm{VF}(\theta, g)$ by employing the optimal $g^*$ during computation.

$$\mathcal{L}_{f,\beta}(\theta) = -\log \sigma\big(\beta(\mathrm{VF}(\theta, g^*))\big).$$

Under the above assumption, the expression Eqn. (8) can be rewritten as:

$$\widetilde{\mathrm{VF}}_{JS}(\theta, g^*) = \mathrm{VF}_{JS}(\theta, g^*) + \mathrm{Bias}_{JS}(\theta, g^*)$$

$$\mathrm{Bias}_{JS}(\theta, g^*)$$

$$= \triangle_{JS}^{y_\ell}(\theta, g^*) - \triangle_{JS}^{y_w}(\theta, g^*)$$

$$= e_f \Big[ \mathbb{E}_{Z_\ell \sim D_\ell}[g^*(Z_\ell)] - \mathbb{E}_{Z_w \sim D_w}[f^*(g^*(Z_w))] \Big] - e_f \Big[ \mathbb{E}_{Z_w \sim D_w}[g^*(Z_w)] - \mathbb{E}_{Z_\ell \sim D_\ell}[f^*(g^*(Z_\ell))] \Big]$$

$$= e_f \sup \mathrm{VF}_{JS}^*(\theta, g) - e_f \sup \mathrm{VF}_{JS}(\theta, g)$$

$$= e_f f_{JS}(D_\ell \| D_w) - e_f f_{JS}(D_w \| D_\ell)$$

As Jenshon-Shannon is inherently symmetric,

$$f_{JS}(D_w \| D_\ell) = \frac{1}{2} D_{KL}(D_w \| M) + \frac{1}{2} D_{KL}(D_l \| M),$$

Where $M = \frac{1}{2}(D_w + D_\ell)$, then

$$\text{Bias}_{JS}(\theta, g^*)$$
$$= e_f f_{JS}(D_\ell \| D_w) - e_f f_{JS}(D_w \| D_\ell)$$
$$= \frac{1}{2} e_f \left[ D_{KL}(D_w \| M) + D_{KL}(D_l \| M) - D_{KL}(D_l \| M) - D_{KL}(D_w \| M) \right]$$
$$= 0$$

Ultimately, the explicit form of JS divergence:

$$\widetilde{\text{VF}}_{JS}(\theta, g^*) = \text{VF}_{JS}(\theta, g^*).$$

We proof the claim.

## C  SUPPLEMENTARY EXPERIMENTS

Table 7: outputs parameters in $f$-DPO

| Benchmark | LLMs | temperature | max-tokens |
|-----------|------|-------------|------------|
| MT-bench | pythia2.8b | 0.7 | 1024 |
| TruthfulQA | llama-2-7b | 0.7 | 256 |

Table 8: Training parameters in $f$-DPO

| dataset | LLMs | learning rate | $\beta$ | Batch Size |
|---------|------|---------------|---------|------------|
| HH-RLHF | pythia2.8b | 5e-7 | 0.1 | 64 |
| Ultrafeedback | llama-2-7b | 5e-7 | 0.1 | 64 |

Table 9: varing noise rate $e_f$ on ultrafeedback

| **Preference Accuracy** | $e_f = 0.1$ | $e_f = 0.2$ | $e_f = 0.3$ | $e_f = 0.4$ |
|-------------------------|-------------|-------------|-------------|-------------|
| Best Accuracy | 79.30 | 75.00 | 70.70 | 65.63 |
| Last Accuracy | 79.30 | 75.00 | 70.31 | 62.89 |

## D  COMPARISON OF GENERATED SAMPLES

Table 10: $f$-DPO outperformed DPO, which was trained on Llama-2-7B, while judging by GPT-4.

| Prompt | What is some cool music from the 1920s? |
|---|---|
| $f$-DPO | The 1920s was a significant period in music history, with several genres emerging and gaining popularity. Here are some cool music from the 1920s: 1. Jazz: The 1920s saw the birth of jazz, a genre that combined African American musical traditions with European harmony and improvisation. Some notable jazz artists from this period include Louis Armstrong, Duke Ellington, and Bessie Smith. 2. Blues: The blues, a genre that originated in the Deep South, also gained popularity during the 1920s. Artists such as Robert Johnson, Bukka White, and Charley Patton helped to popularize the blues. 3. Swing: Swing music, a style that emerged from jazz, became popular in the 1920s. Artists such as Duke Ellington, Count Basie, and Fletcher Henderson helped to popularize swing. 4. Pop music: Pop music, influenced by jazz and blues, also gained popularity in the 1920s. Artists such as Irving Berlin, Cole Porter, and George Gershwin helped to popularize pop music. 5. Tin Pan Alley: Tin Pan Alley, a term for the music publishing industry in New York City, emerged in the 1920s. Artists like Irving Berlin, George Gershwin, and Cole Porter helped popularize this music. 6. Vaudeville: Vaudeville, a type of variety entertainment popular in the 1920s, also influenced music. Artists such as Al Jolson, Sophie Tucker, and Ethel Merman helped to popularize vaudeville music. 7. Broadway: Broadway musicals influenced music during the 1920s. Artists such as Cole Porter, George Gershwin, and Irving Berlin helped popularize Broadway music. 8. The Great Gatsby: The novel "The Great Gatsby" by F. Scott Fitzgerald also influenced music during this period. Artists such as Louis Armstrong, Duke Ellington, and Bessie Smith helped popularize the music of the 1920s. |
| DPO | The 1920s, also known as the Roaring Twenties, was a time of great cultural and artistic innovation, particularly in the realm of music. Here are some cool and influential songs from the 1920s that are still enjoyed today: 1. "Livery Stable Blues" by Bessie Smith: Released in 1923, this song is considered a classic of the blues genre and is known for its powerful vocals and emotional delivery. 2. "My Happiness" by Harry Woods: First recorded by the Boswell Sisters in 1931, but the original version was released in 1927 by Harry Woods. Its lively and catchy melody has made it popular over the years. 3. "Some of These Days" by Sophie Tucker: Released in 1924, this jazz standard has been covered by many artists, including Louis Armstrong and Ella Fitzgerald. 4. "I'm Just Wild About Harry" by Jimmy Durante: Written by Bert Kalmar and Harry Ruby in 1921, first performed by Durante in a Broadway musical. Catchy tune and playful lyrics. 5. "Yes! We Have No Bananas" by Frank Silver and Irving Cohn: Written in 1923, became a huge hit with catchy melody and humorous lyrics. 6. "Makin' Whoopee!" by Eddie Cantor: Released in 1928, classic of jazz and swing genres, upbeat tempo and playful lyrics make it fun to listen. |