# OpenReview forum: "Robust Direct Preference Optimization via Variational form f-divergence"
_ICLR.cc/2026/Conference — ICLR 2026 Conference Withdrawn Submission_

### Official Review · Reviewer_drKW · 2025-10-25

**Soundness:** 1
**Presentation:** 2
**Contribution:** 2
**Rating:** 2
**Confidence:** 4

**Summary:**

Noisy annotations in preference data can lead to learning undesirable behavior. Current methods such as Robust DPO mitigate such effects but require knowing the noise rate and being able to distinguish noisy vs. clean samples. Dr.DPO is another method which does not require noise rate estimation and is effective on both clean and noisy datasets. The authors introduce maximizing the f-divergence between preferred and unpreferred distributions and use this formulation to modify the loss function. They demonstrate improvements compared to baseline methods.

**Strengths:**

Novelty - The paper introduces an original perspective that aims to robustly learn from preference data by framing the objective as f-divergence maximization between the preferred and unpreferred distributions. Under this setup, the authors demonstrate that different f-divergences have varying behavior for noisy data and show that the Jensen-Shannon divergence is invariant to preference noise.

Empirical Performance - The experiments demonstrate that the method can perform better than existing methods on multiple tasks suggesting the method may lead to more robust preference learning. The tasks include both preference accuracy and generation-based metrics on MTBench.

**Weaknesses:**

Experimental Setup - The experimental setup could be more consistent across evaluations and provide further details. In particular, while evaluations on MT-Bench are based on Pythia models trained on HH-RLHF while the evaluations on TruthfulQA are based on LLaMa-2 models trained on UltraFeedback. The experiments would be significantly more informative if all evaluations were based on the same starting model trained on the same dataset. One useful detail that could be provided is the preference accuracy/downstream performance of methods when the noise rate is 0 so that for each instance, there is a clear starting point that allows for a straightforward comparison of how much the performance drops once noise is introduced. Based on the current results, the generality of the method’s benefits seems unclear.

Technical Clarity - The technical ideas presented pass over significant details and also contains incorrect statements. In particular, parts of the derivation of the method and modified loss function should be further discussed. For example, in lines 196-198, it is unclear what is meant by the variational representation being an empirical estimate in contrast to a strict equivalence and why this is a reasonable choice to make. Some technical details that require correcting or are missing:
- Lines 31-32: The reward model itself is not trained with PPO.
- Lines 149-151: The Z’s are not defined.
- Lines 191-192: Minimizing the DPO loss is not equivalent to point-wise maximizing the implicit reward and furthermore, the DPO loss is derived from reward maximization with KL regularization used to limit the extent to which reward is maximized.
- Lines 202-203: D_l is not defined.

Impact - The work aims to build on robust preference learning methods and in particular mentions Dr.DPO as an effective method that does not require noise rate estimation but lacks rigorous theoretical justification regarding its robustness. However, while the work demonstrates some improvements over Dr.DPO, the improvements are shown on limited settings and the work does not provide a rigorous theoretical justification for the objective. In particular, while the works demonstrates that the f-divergence itself is robust to noise, the work does not address properties of the final resulting objective such as explaining the choice of log-sigmoid, providing a gradient analysis, or deriving its optimal policy.

Actionable changes:
- Providing further explanation on the derivation of the final objective and its properties
- Clarifying definitions and steps in the derivation and updating incorrect statements
- Providing experimental results under a consistent setup with the same base model and dataset
- For results across noise rates, providing performance at a noise rate of 0 for a clear baseline

**Questions:**

Questions:
- Could you explain the choice of having the objective be negative log sigmoid of beta times VF term?
- A follow up question I have is I'm not certain that maximizing the f-divergence term necessarily means learning correct preferences. Couldn't it also be maximized by learning incorrect preferences? I would appreciate some clarification here as well.

Suggestions are listed in weaknesses.

---

### Official Review · Reviewer_3Kse · 2025-10-30

**Soundness:** 3
**Presentation:** 2
**Contribution:** 3
**Rating:** 4
**Confidence:** 3

**Summary:**

This paper attempts to examine how the properties of f-divergence can improve the performance of DPO with noisy labels. The authors present a theoretical analysis of a DPO like loss function based on f-divergence. They analyze the properties of the variational form of the loss function and show that the Total Variation form of f-divergence is useful when the noise rate is known while the JS divergence is invariant to noise. In empirical studies, they show that the reformulated loss function based on DPO performs better on HH-RLHF and UltraFeedback datasets in comparison to several strong baselines

**Strengths:**

1. This paper presents a novel analysis - reformulating the DPO loss function based on f-divergence, it's theoretical properties under noisy conditions.
2. The experiments show that this loss function shows better performance over other baselines at diverse noise levels

**Weaknesses:**

1. There is room for improvement in presentation of the theoretical proofs. Particularly in sections 2 and 4, the equations are not numbered and it is difficult to follow how different equations connect to one another. Furthermore, there is no intuition or analytical understandings given which explains why the reformulated loss function shows better performance in comparison to the baselines
2. The authors do not discuss the additional computational needs from variational optimization which is not needed in the case of the baselines
3. The experiments are limited to English-only text tasks with mid-sized models. Since different loss functions are being examined with respect to noisy labels, experiments with larger models and other tasks will solidify the paper.

**Questions:**

Please see the weaknesses.

---

### Official Review · Reviewer_5ZUj · 2025-11-01

**Soundness:** 2
**Presentation:** 3
**Contribution:** 1
**Rating:** 4
**Confidence:** 4

**Summary:**

This work studies on direct preference optimization and proposes a novel strategy with the introduction of f-divergence.

While the topic is important and the paper presents a method that is conceptually reasonable, I have significant concerns on the novelty, motivation and the experimental evaluation. As a result, I assign a score of 4.

**Strengths:**

1.	This work studies on an important problem.
2.	The proposed method appears to be reasonable.
3.	Extensive experiments on real-world datasets have been conducted to verify the efficacy of the proposed method.

**Weaknesses:**

1.	My major concern lies on the novelty. The use of f-divergence in DPO has already been extensively explored in recent studies, like [a1][a2]. This work does not adequately review these contributions, nor does it clearly explain how the proposed method differs from them.

2.	My another concern lies on the unclear motivation: 1) While the paper claims benefits from introducing f-divergence, these benefits appear to arise mainly from the use of JS-divergence, which is also the only form implemented in the experiments. 2) Existing denoising DPO strategies are not sufficiently compared, and it remains unclear why and how the proposed method would outperform them. 3) The theoretical analysis relies heavily on the assumption that e_w$=$e_l$,  whose validity should be justified.


3.	There are also some concerns on the experiments: 1) Some important baselines like [a1][a2] are missing  from the comparisons. 2) Experiments are conducted only on a relatively small LLM, Pythia-2.8B. Testing on larger and more diverse models would better demonstrate scalability and effectiveness. 3) Evaluation is limited to two benchmarks. Incorporating additional datasets would strengthen the empirical evidence.

[a1] AISTATS’25: f-PO: Generalizing Preference Optimization with f-divergence Minimization
[a2] iclr’25: beyond Reverse KL: Generalizing Direct Preference Optimization with Diverse Divergence Constraints

**Questions:**

Please refer to weaknesses.

---

### Official Review · Reviewer_ogr5 · 2025-11-01

**Soundness:** 2
**Presentation:** 2
**Contribution:** 1
**Rating:** 2
**Confidence:** 4

**Summary:**

This paper use f-DPO loss function (which is already accepted in ICLR 2024 [1]), to handles noisy human preference annotations when aligning Large Language Models. Standard DPO suffers from noisily annotated human preferences, and existing robust approaches either require knowledge of the noise transition rates or need additional models for correction.
The paper investigates when f-divergence is immune to imperfect preference annotations by maximizing f-divergence between noisy preferred and unpreferred data distributions. When noise ratio is known, the Total Variation formulation can be transformed to represent the clean dataset with a multiplicative factor (1-2ef). The Jensen-Shannon formulation is invariant to noise - it yields identical results under both noisy and clean preferences, even without knowing the noise rate
Experiments on HH-RLHF and UltraFeedback datasets with MT-Bench and TruthfulQA benchmarks show that the JS variational form: 1) Enhances ability to generate preferred responses under noisy conditions; 2) Improves factual accuracy of outputs; 3. Consistently outperforms baselines (DPO, cDPO, rDPO, IPO, Dr.DPO) across various noise levels.

[1] Beyond Reverse KL: Generalizing Direct Preference Optimization with Diverse Divergence Constraints.

**Strengths:**

1. The paper makes a genuine theoretical contribution by analyzing the behavior of f-divergence variational forms under noisy preference data.
2. Noisy human preference annotations are a real and costly problem in RLHF. The paper's finding that JS divergence naturally provides noise robustness without requiring noise rate estimation or auxiliary models is practically significant.

**Weaknesses:**

## 1. Insufficient Credit to Wang et al. (2023)

**Problem:** The paper adopts the f-divergence variational framework from Wang et al. (2023) but provides inadequate attribution.

**Evidence of Equivalence:**
- **Robust DPO (this paper):** $\mathcal{L}_{JS}(\theta) = -\log \sigma(\beta \cdot VF_{JS}(\theta, g^*))$
- **Beyond Reverse KL (Wang et al.):** $\mathcal{L}(\theta) = -\log\sigma\left(\beta f'\left(\frac{\pi_\theta(y_w|x)}{\pi_{ref}(y_w|x)}\right) - \beta f'\left(\frac{\pi_\theta(y_l|x)}{\pi_{ref}(y_l|x)}\right)\right)$
- **Proof:** $g^*(v) = f'(e^v) = \log \frac{2e^v}{1+e^v}$ → **Same loss function, different notation**

## 2.  Undefined Symbols
- Line 117: $\beta, \pi, \theta$ not defined
- Line 122: $\beta$ definition missing; $r(x,y)$ needs explanation
- Line 144: Clarify if $x$ is the same as Line 111

## 3. Errors and Typos
- Line 118: $(x, y)$ should be $(y_w, y_l)$
- Line 119: "SFT policy" should be "reference policy"
- Line 183: $\log\sigma$ is informal notation

**Questions:**

See Weaknesses.

---

### Note · Authors · 2026-01-05

I have read and agree with the venue's withdrawal policy on behalf of myself and my co-authors.